# SURFACE-BASED PEPTIDE DESIGN WITH MULTI-MODAL FLOW MATCHING

## ABSTRACT

Therapeutic peptides show promise in targeting previously undruggable binding sites, with recent advancements in deep generative models enabling full-atom peptide co-design for specific protein receptors. However, the critical role of molecular surfaces in protein-protein interactions (PPIs) has been underexplored. To bridge this gap, we propose an *omni-design* peptides generation paradigm, called SurfFlow, a novel surface-based generative algorithm that enables comprehensive co-design of sequence, structure, and surface for peptides. SurfFlow employs a multi-modality conditional flow matching (CFM) architecture to learn distributions of surface geometries and biochemical properties, enhancing peptide binding accuracy. Evaluated on the comprehensive PepMerge benchmark, SurfFlow consistently outperforms full-atom baselines across all metrics. These results highlight the advantages of considering molecular surfaces in *de novo* peptide discovery and demonstrate the potential of integrating multiple protein modalities for more effective therapeutic peptide discovery. Anonymous codes are available at `https://anonymous.4open.science/r/SurfFlow-880B/`.

## 1 INTRODUCTION

Peptides, short-chain proteins composed of roughly 2 to 50 amino acids linked by peptide bonds, play critical roles in various biological processes, including cell signaling, enzymatic catalysis, and immune responses (Wang et al., 2022). They are essential mediators in pharmacology due to their ability to bind cell surface receptors with high affinity and specificity, inducing intracellular effects with low toxicity, minimal immunogenicity, and ease of delivery, as well as being readily synthesized and modified. (Muttenthaler et al., 2021). Conventional simulation or searching methods of peptide discovery rely on frequent calculations of physical energy functions, a process hindered by the vast design space of peptides (Bhardwaj et al., 2016). This has spurred a growing demand for computational approaches that facilitate *in silico* peptide design and analysis.

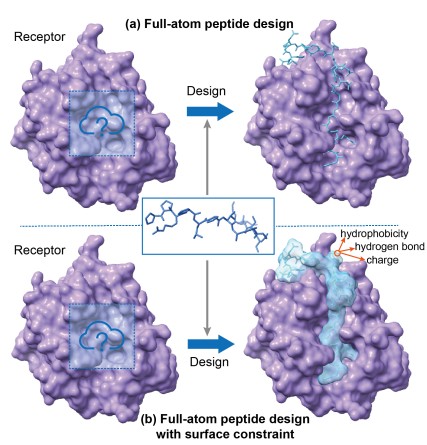

Figure 1: Comparison of full-atom peptide design with and without the surface constraint.

Recent progress in diffusion probabilistic (Ho et al., 2020) and flow-based (Lipman et al., 2022) models have illuminated tremendous promise in molecular design (Guan et al., 2023), antibody engineering (Luo et al., 2022; Martinkus et al., 2024), *de novo* protein design (Yim et al., 2023; Wu et al., 2024) as well as peptide discovery (Ramasubramanian et al., 2024). Peptides, when unbound, often exist in high-energy, high-entropy states with unstable conformations, only becoming functional upon binding to target receptors. Therefore, peptide design must be explicitly conditioned on binding pockets Vanhee et al. (2011). Besides, as residues interact with each other through non-covalent forces formed by side-chain groups, increasing efforts have been made to capture protein-peptide interactions through full-atom geometries (Kong et al., 2024; Lin et al., 2024; Li et al., 2024), pushing sequence-structure co-design beyond just backbones.

Despite advances, growing attention is given to molecular surfaces in protein-protein interactions (PPIs), as these interactions are largely dictated by how complementarily the surfaces of interacting proteins fit together (Kastritis & Bonvin, 2013; Song et al., 2024). The shape, electrostatic potential, and hydrophobicity of molecular surfaces are key determinants of the interaction's strength and specificity (Jones & Thornton, 1996; Lee et al., 2023; Wu & Li, 2024b). The surface geometry such as protrusions, grooves, and clefts enables lock-and-key or induced-fit mechanisms essential for specific binding. The molecular surfaces act as a fundamental interface that dictates how proteins recognize and bind to each other, making it as essential as side chains in PPIs. For these reasons, it is vital to simultaneously consider all molecular modalities – sequence, structure, and surface – during peptide generation (see Fig. 1), enhancing the consistency across aspects in what we term *omni-design*.

Toward this goal, we propose SurfFlow, a surface-based generative algorithm built upon a multi-modality Conditional Flow Matching (CFM) architecture (Lipman et al., 2022; Albergo & Vanden-Eijnden, 2022; Albergo et al., 2023). In SurfFlow, CFM is applied not only to internal geometries such as backbone and side-chain angles but also to the molecular surface, represented by surface point positions and unit norm vectors as a rigid frame in SE(3). Additionally, omni-design requires incorporating biochemical property constraints, as complementary surface geometries alone do not guarantee successful binding-accurate placement of charges, polarity, or hydrophobicity at the binding interface is also necessary (Gainza et al., 2023). To achieve this, SurfFlow learns the transformation from prior distributions to the expected biochemical property distributions. Since surface features such as hydrophobicity are categorical, we apply the Discrete Flow Models (DFMs) (Campbell et al., 2024) to discrete data space using Continuous-Time Markov Chains (CTMC). Finally, recognizing that key peptide characteristics like cyclicity and disulfide bonds influence stability and binding affinity(Buckton et al., 2021), we include these factors as additional conditions to enhance the capacity and generalization of SurfFlow. We evaluate SurfFlow on the comprehensive peptide design benchmark PepMerge (Li et al., 2024), and experiments demonstrate that it consistently outperforms full-atom baselines across all metrics, highlighting the advantages of considering surfaces for *de novo* peptide design.

## 2 PRELIMINARY AND BACKGROUND

**Proteins and Molecular Surfaces.** A protein is a biomolecule consisting of multiple amino acid residues, each defined by its type, backbone frame, and side-chain torsion angles (Fisher, 2001). The type of the $i$-th residue, denoted by $a_i \in \{1 \dots 20\}$, is determined by its side-chain R group. The rigid frame of each residue is constructed from the coordinates of four backbone heavy atoms N, C$\alpha$, C, and O, with C$\alpha$ positioned at the origin. This frame is represented by a position vector $\boldsymbol{x}_i \in \mathbb{R}^3$ and a rotation matrix $O_i \in \mathrm{SO}(3)$ (Jumper et al., 2021). Unlike the backbone, the side-chain conformation is more flexible, involving up to four rotatable torsion angles between side-chain atoms, denoted by $\boldsymbol{\chi}_i \in [0, 2\pi)^4$. Additionally, the backbone torsion angle $\varphi_i \in [0, 2\pi)$ affects the position of the oxygen atom.

We further consider the molecular surface, which is computed by moving a probe of a certain radius (approximately 1 Å) along the protein to calculate the Solvent Accessible Surface (SAS) and Solvent Excluded Surface (SES). The probe's coordinates define the surface as an oriented point cloud $Q = \{q_i : 1 \le i \le m\}$. Each surface point $q_i$ has associated attributes $(\boldsymbol{x}_i^s, \boldsymbol{n}_i^s, \boldsymbol{\tau}_i^s, \boldsymbol{\Upsilon}_i^s)$, where $\boldsymbol{x}_i^s \in \mathbb{R}^3$ represents its 3D coordinates, and $\boldsymbol{n}_i^s \in \mathbb{R}^3$ is the corresponding unit normal vector. $\boldsymbol{\tau}_i^s \in \mathbb{R}^{\psi_\tau}$ and $\boldsymbol{\Upsilon}_i^s \in \mathbb{R}^{\psi_\Upsilon}$ capture its continuous and categorical physicochemical properties, such as hydrophobicity, hydrogen bonding, and charge (Gainza et al., 2020; Song et al., 2024; Wu & Li, 2024b).

This work focuses on designing peptides based on target proteins. Formally, given a peptide $C^{\mathrm{pep}}$ with $n_{\mathrm{pep}}$ residues and a target protein $C^{\mathrm{rec}}$ with $n_{\mathrm{rec}}$ residues, we aim to model the conditional joint distribution $p\left(C^{\mathrm{pep}} \mid C^{\mathrm{rec}}\right)$. The receptor can be sufficiently and succinctly parameterized as $C^{\mathrm{rec}} = \{(a_i, O_i, \boldsymbol{x}_i, \boldsymbol{\chi}_i)\}_{i=1}^{n_{\mathrm{rec}}}$, where $\boldsymbol{\chi}_i[0] = \varphi_i$ and $\boldsymbol{\chi}_i \in [0, 2\pi)^5$. As for the ligand peptide, the surface is also portrayed, resulting in $C_{\mathrm{pep}} = \{(a_j, O_j, \boldsymbol{x}_j, \boldsymbol{\chi}_j)\}_{j=1}^{n_{\mathrm{pep}}} \cup \{(\boldsymbol{x}_i^s, \boldsymbol{n}_i^s, \boldsymbol{\tau}_i^s, \boldsymbol{\Upsilon}_i^s)\}_{i=1}^{m}$ with $m \gg n_{\mathrm{pep}}$. Practically, software like PyMol (DeLano et al., 2002) or MSMS (Robinson et al., 2014) can be utilized to compute the raw molecular surface of a protein.

**Probability Path and Flow.** Let $\mathbb{P}(\mathcal{M})$ be the space of probability distributions on a manifold $\mathcal{M}$ with a Riemanian metric $g$. A probability path $p_t : [0,1] \times \mathcal{M} \to \mathbb{P}(\mathcal{M})$ is an interpolation in the probability space between two distributions $p_0, p_1 \in \mathbb{P}(\mathcal{M})$ indexed by a continuous parameter $t$.

A flow on $\mathcal{M}$ is defined by a one-parameter diffeomorphism $\Phi : \mathcal{M} \to \mathcal{M}$, which is the result of integrating instantaneous deformations described by a time-varying vector field $u_t \in \mathcal{U}$. $u_t(x) \in \mathcal{T}_x\mathcal{M}$ is the gradient vector of the path $p_t$ on $x$ at time $t$. By solving the following Ordinary Differential Equation (ODE) on $\mathcal{M}$ over $t \in [0,1]$ with an initial condition of $\phi_0(x) = x$: $\frac{\mathrm{d}\phi_t}{\mathrm{d}t}(x) = u_t(\phi_t(x))$. We acquire the time-dependent flow $\phi_t : \mathcal{M} \to \mathcal{M}$ and the final diffeomorphism by setting $\Phi(x) = \phi_1(x)$. Notably, $\phi_t(x)$ is also the solution of another ODE: $\mathrm{d}x = u_t(x)\mathrm{d}t$, which transports the point $x$ along the vector field $u_t(x)$ from time 0 up to time $t$. Given a source density $p_0$, $\phi_t(x)$ induces a push-forward operation $p_t = [\phi_t]_\# p_0$. It reshapes the point density $x \sim p_0$ to a more complicated one along $u_t(x)$, and the change-of-variable operator $\#$ is defined by $[\phi_t]_\# p_0(x) = p_0\left(\phi_t^{-1}(x)\right) \det\left[\frac{\partial \phi_t^{-1}}{\partial x}(x)\right]$. The time-varying density $p_t$ is characterized by the Fokker-Planck equation: $\frac{\mathrm{d}p_t}{\mathrm{d}t} = -\mathrm{div}(u_t p_t)$, also known as the continuity equation. Under these conditions, $u_t$ is said to be the probability flow for $p_t$, and $p_t$ is said to be the marginal probability path generated by $u_t$. Flow Matching (FM) (Lipman et al., 2022; Albergo & Vanden-Eijnden, 2022; Albergo et al., 2023) trains a Continuous Normalizing Flow (CNF) by fitting a vector field $v_\theta \in \mathcal{U}$ with parameters $\theta$ to a target vector field $u_t$ that produce a probability path $p_t$. Its objective falls at the tangent space as:

$$\mathcal{L}_{\mathrm{RFM}}(\theta) = \mathbb{E}_{t \sim \mathcal{U}[0,1], x \sim p_t(x)} \|v_\theta(x,t) - u_t(x)\|_g^2, \tag{1}$$

As $u_t$ is intractable, an alternative is to construct the conditional probability path $p_t(x|x_1)$ with a conditional vector field $u_t(x|x_1)$. The objective becomes: $\mathcal{L}_{\mathrm{CRFM}}(\theta) = \mathbb{E}_{t \sim \mathcal{U}[0,1], x_1 \sim p_1(x_1), x \sim p_t(x|x_1)} \|v_\theta(x,t) - u_t(x|x_1)\|_g^2$. Riemannian FM and Conditional Riemannian FM (CRFM) objectives are proven to share the same gradients (Lipman et al., 2022; Tong et al., 2023; Chen & Lipman, 2023). During inference, one can solve the ODE related to the neural vector field $v_\theta$ to push $x_0 \in \mathcal{M}$ from the source distribution $p_0$ to the data distribution $p_1$ in time.

**Continuous-Time Markov Chains.** CTMC (Norris, 1998) is a class of continuous-time discrete stochastic processes and is closely linked to probability flows. Suppose a categorical variable $x$ has $S$ states and its trajectory $x_t$ over time $t \in [0,1]$ follows a CTMC, $x_t$ alternates between resting in its current state and periodically jumping to another randomly chosen state. The frequency and destination of the jumps are determined by the rate matrix $R_t \in \mathbb{R}^{S \times S}$ with the constraint its off-diagonal elements are non-negative. The probability $x_t$ will jump to a different state $j$ is $R_t(x_t, j)\,\mathrm{d}t$ for the next infinitesimal time step $\mathrm{d}t$. The transition probability is written as:

$$p_{t+\mathrm{d}t|t}(j \mid x_t) = \begin{cases} R_t(x_t, j)\,\mathrm{d}t & \text{for } j \neq x_t \\ 1 + R_t(x_t, x_t)\,\mathrm{d}t & \text{for } j = x_t \end{cases} = \delta\{x_t, j\} + R_t(x_t, j)\,\mathrm{d}t, \tag{2}$$

where $\delta\{i, j\}$ is the Kronecker delta. $\delta\{i, j\}$ is 1 when $i = j$ and is otherwise 0. $R_t(x_t, x_t) := -\sum_{k \neq x_t} R_t(x_t, k)$ in order for $p_{t+\mathrm{d}t|t}(\cdot \mid i)$ to sum to 1. Using compact notation, $p_{t+\mathrm{d}t|t}$ is therefore a categorical distribution with probabilities $\delta\{x_t, \cdot\} + R_t(x_t, \cdot)\,\mathrm{d}t$ denoted as $\mathrm{Cat}(\delta\{x_t, j\} + R_t(x_t, j)\,\mathrm{d}t)$. Namely, $j \sim p_{t+\mathrm{d}t|t}(j \mid x_t) \iff j \sim \mathrm{Cat}(\delta\{x_t, j\} + R_t(x_t, j)\,\mathrm{d}t)$. In practice, we need to simulate the sequence trajectory with finite time intervals $\Delta t$. A trajectory can be simulated with Euler steps (Sun et al., 2022).

$$x_{t+\Delta t} \sim \mathrm{Cat}(\delta\{x_t, x_{t+\Delta t}\} + R_t(x_t, x_{t+\Delta t})\,\Delta t), \tag{3}$$

where the variable $x$ starts from an initial sample $x_0 \sim p_0$ at time $t = 0$. The rate matrix $R_t$ along with an initial distribution $p_0$ together define the CTMC. With the marginal distribution at time $t$ as $p_t(x_t)$, the Kolmogorov equation allows us to relate the rate matrix $R_t$ to the change in $p_t(x_t)$:

$$\partial_t p_t(x_t) = \underbrace{\sum_{j \neq x_t} R_t(j, x_t)\,p_t(j)}_{\text{incoming}} - \underbrace{\sum_{j \neq x_t} R_t(x_t, j)\,p_t(x_t)}_{\text{outgoing}}. \tag{4}$$

The difference between the incoming and outgoing probability mass is the time derivative of the marginal $\partial_t p_t(x_t)$. Subsequently, we attain $\partial_t p_t = R_t^\top p_t$ where the marginals are treated as probability mass vectors: $p_t \in [0,1]^S$ and defines an ODE in a vector space. The probability path $p_t$ is said to be generated by $R_t$ if $\partial_t p_t = R_t^\top p_t$ for $\forall t \in [0,1]$ (Campbell et al., 2024).

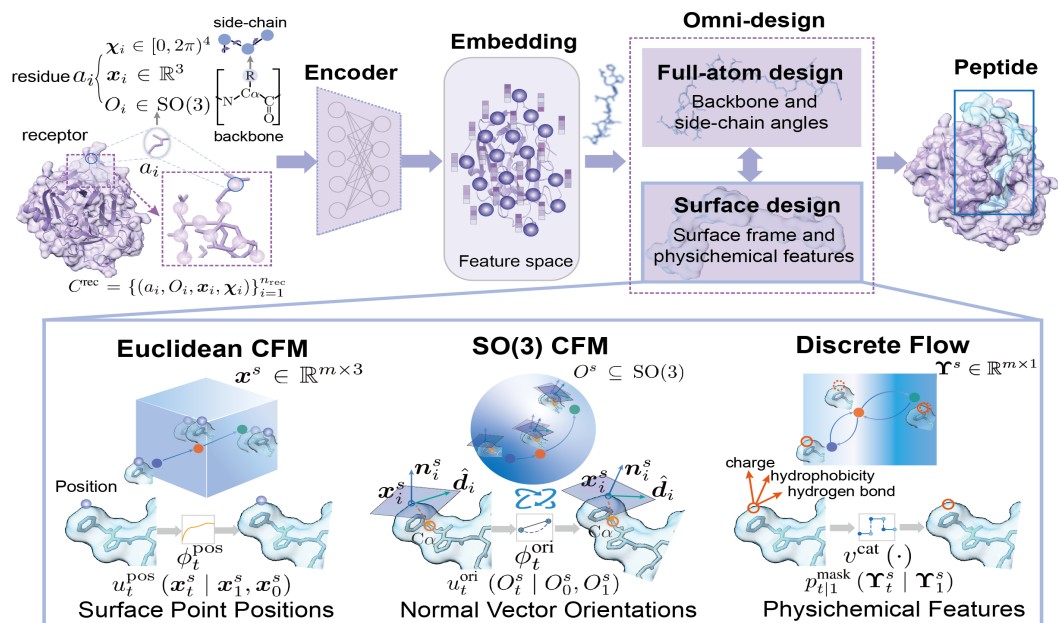

Figure 2: Workflow of SurfFlow for our peptide omni-design, which considers the multi-modality consistency among sequence, structure, and molecular surface during the generation process.

## 3 METHOD

A molecular surface encapsulates both the 3D geometry of a protein in Euclidean space and its biochemical attributes, such as hydrophobicity and charge (Gainza et al., 2020). The interplay between surface shape and these biochemical properties is essential for defining a protein's function. Given a target receptor with specific geometric and biochemical constraints, SurfFlow concurrently generates the peptide's internal structure and external surface. Moreover, it can also account for key factors such as cyclicity and disulfide bonds (see Fig. 2).

### 3.1 FLOW MATCHING FOR SURFACE GENERATION

A CFM framework is employed to learn the conditional peptide distribution based on its target protein $p\left(C^{\text{pep}} \mid C^{\text{rec}}\right)$. This joint probability is empirically decomposed into the product of probabilities of the internal structure elements and the external surface elements:

$$p\left(C^{\text{pep}} \mid C^{\text{rec}}\right) \propto p\left(\{(a_j, O_j, \boldsymbol{x}_j, \boldsymbol{\chi}_j)\}_{j=1}^{n_{\text{pep}}} \mid C^{\text{rec}}\right) p\left(\{(\boldsymbol{x}_i^s, \boldsymbol{n}_i^s, \boldsymbol{\tau}_i^s, \boldsymbol{\Upsilon}_i^s)\}_{i=1}^m \mid C^{\text{rec}}\right), \quad (5)$$

where $p\left(\{(\boldsymbol{x}_i^s, \boldsymbol{n}_i^s, \boldsymbol{\tau}_i^s, \boldsymbol{\Upsilon}_i^s)\}_{i=1}^m \mid C^{\text{rec}}\right)$ is further separated as the product of four basic elements $p\left(\{\boldsymbol{x}_i^s\}_{i=1}^m \mid C^{\text{rec}}\right) p\left(\{\boldsymbol{n}_i^s\}_{i=1}^m \mid C^{\text{rec}}\right) p\left(\{\boldsymbol{\tau}_i^s\}_{i=1}^m \mid C^{\text{rec}}\right) p\left(\{\boldsymbol{\Upsilon}_i^s\}_{i=1}^m \mid C^{\text{rec}}\right)$. The construction of different probability flows on the surface point's position $p\left(\boldsymbol{x}_i^s \mid C^{\text{rec}}\right)$, orientation $p\left(\boldsymbol{n}_i^s \mid C^{\text{rec}}\right)$, continuous properties $p\left(\boldsymbol{\tau}_i^s \mid C^{\text{rec}}\right)$, and categorical properties $p\left(\boldsymbol{\Upsilon}_i^s \mid C^{\text{rec}}\right)$ is elaborated as follows.

**Position.** Euclidean CFM is utilized to generate surface point positions $\boldsymbol{x}^s \in \mathbb{R}^{m \times 3}$. Following common practice (Yim et al., 2023; Lin et al., 2024; Li et al., 2024), we adopt the standard isotropic Gaussian $\mathcal{N}(0, \boldsymbol{I}_3)$ as the prior, with the target distribution being $p\left(\boldsymbol{x}^s \mid C^{\text{rec}}\right)$. The conditional flow is defined as a linear interpolation between sampled noise $\boldsymbol{x}_0^s \sim \mathcal{N}(0, \boldsymbol{I}_3)$ and data points $\boldsymbol{x}_1^s \sim p\left(\boldsymbol{x}^s \mid C^{\text{rec}}\right)$. This linear interpolation ensures a straight trajectory, contributing to training and sampling efficiency by following the shortest path between two points in Euclidean space (Liu et al., 2022). The conditional vector field $u_t^{\text{pos}}$ is obtained by taking the time derivative of the linear flow $\phi_t^{\text{pos}}$ using Independent Coupling techniques:

$$\phi_t^{\text{pos}}\left(\boldsymbol{x}_0^s, \boldsymbol{x}_1^s\right) = t\boldsymbol{x}_1^s + (1-t)\boldsymbol{x}_0^s, \quad (6)$$

$$u_t^{\text{pos}}\left(\boldsymbol{x}_t^s \mid \boldsymbol{x}_1^s, \boldsymbol{x}_0^s\right) = \boldsymbol{x}_1^s - \boldsymbol{x}_0^s = \frac{\boldsymbol{x}_1^s - \boldsymbol{x}_t^s}{1-t}. \quad (7)$$

We use a time-dependent translation-invariant surface network $v^{\text{pos}}(\cdot)$ to predict the conditional vector field based on the current interpolant $\boldsymbol{x}_t$ and the timestep $t$. The CFM objective of the surface point cloud position is formulated as:

$$\mathcal{L}_{\text{pos}}(\theta) = \mathbb{E}_{t \sim \mathcal{U}(0,1), p(\boldsymbol{x}_1^s), p(\boldsymbol{x}_0^s), p(\boldsymbol{x}_t^s | \boldsymbol{x}_0^s, \boldsymbol{x}_1^s)} \left\| v^{\text{pos}} \left( \boldsymbol{x}_t^s, t, C^{\text{rec}} \right) - \left( \boldsymbol{x}_1^s - \boldsymbol{x}_0^s \right) \right\|_2^2, \quad (8)$$

where $\mathcal{U}(0,1)$ is a uniform distribution on $[0,1]$. During generation, we first sample from the prior $\boldsymbol{x}_0^s \sim \mathcal{N}(0, \boldsymbol{I}_3)$ and solve the probability flow with the learned predictor $v^{\text{pos}}(\cdot)$ using the $N$-step forward Euler method to get the position of residue $j$ with $t = \left\{ 0, \ldots, \frac{N-1}{N} \right\}$:

$$\boldsymbol{x}_{t + \frac{1}{N}}^s = \boldsymbol{x}_t^s + \frac{1}{N} v^{\text{pos}} \left( \boldsymbol{x}_t^s, t, C^{\text{rec}} \right). \quad (9)$$

**Normal Vector Orientations.** The normal vector $\boldsymbol{n}_i^s$ is a unit vector perpendicular to the tangent plane of the surface at $\boldsymbol{x}_i^s$. It reveals essential geometric information of the surface orientation and curvature, directly tied to the protein's functional (Song et al., 2024; Wu & Li, 2024b). Convex regions, for example, might be more accessible for binding, while concave regions might be better suited for pockets or clefts involved in substrate binding (Laskowski et al., 1996). To capture its orientation, we construct a set of rotation matrices $O^s \subseteq \text{SO}(3)$ with respect to the global frame, whose element is defined by $O_i^s = \left( \boldsymbol{n}_i^s, \hat{\boldsymbol{d}}_i, \boldsymbol{n}_i^s \times \hat{\boldsymbol{d}}_i \right) \in \text{SO}(3)$. Here, $\hat{\boldsymbol{d}}_i$ is a unit vector orthogonal to $\boldsymbol{n}_i^s$ and is acquired by normalizing the cross product between $\boldsymbol{n}_i^s$ and the direction pointing from the surface point $\boldsymbol{x}_i^s$ to its nearest $C_\alpha$ coordinate. This frame $O_i^s$ effectively describes the geometric relationship between the protein surface and the underlying backbone structure.

The 3D rotation group $\text{SO}(3)$ is a smooth Riemannian manifold, with its tangent space, $\mathfrak{so}(3)$, forming a Lie algebra consisting of skew-symmetric matrices. Elements of $\mathfrak{so}(3)$ can be viewed as infinitesimal rotations around specific axes and represented as rotation vectors in $\mathbb{R}^3$ (Blanco-Claraco, 2021). In line with prior work (Li et al., 2024; Lin et al., 2024), we adopt the uniform distribution over $\text{SO}(3)$ as the prior $p\left( O_s^0 \right)$. Just as FM in Euclidean space is based on the shortest path between two points, we extend this idea to $\text{SO}(3)$ by using geodesics, which define the minimal rotational distance between two orientations (Lee, 2018). These geodesics provide a natural framework for interpolating and evolving rotations while respecting the geometry of the manifold (Bose et al., 2023; Yim et al., 2023). The conditional flow $\phi^{\text{ori}}$ and vector field $u_t^{\text{ori}}$ are constructed by geodesic interpolation between $O_0^s \subseteq \mathcal{U}(\text{SO}(3))$ and $O_1^s \in p\left( O^s \mid C^{\text{rec}} \right)$, with the geodesic distance decreasing linearly over time:

$$\phi_t^{\text{ori}} \left( O_0^s, O_1^s \right) = \exp_{O_0^s} \left( t \log_{O_0^s} \left( O_1^s \right) \right), \quad (10)$$

$$u_t^{\text{ori}} \left( O_t^s \mid O_0^s, O_1^s \right) = \frac{\log_{O_t^s} \left( O_1^s \right)}{1 - t}, \quad (11)$$

where $\exp(\cdot)$ and $\log(\cdot)$ are the exponential and logarithm maps on $\text{SO}(3)$ that can be computed efficiently using Rodrigues' formula (Li et al., 2024). A rotation-equivariant surface network $v^{\text{ori}}(\cdot)$ is applied to predict the vector field $u_t^{\text{ori}}$, represented as rotation vectors. The CFM objective on $\text{SO}(3)$ is formulated as:

$$\mathcal{L}_{\text{ori}}(\theta) = \mathbb{E}_{t \sim \mathcal{U}(0,1), p(O_1^s), p(O_0^s), p(O_t^s | O_0^s, O_1^s)} \left\| v^{\text{ori}} \left( O_t^s, t, C^{\text{rec}} \right) - \frac{\log_{O_t^s} \left( O_1^s \right)}{1 - t} \right\|_{\text{SO}(3)}^2, \quad (12)$$

where the vector field $v^{\text{ori}}(\cdot)$ is defined in the tangent space $\mathfrak{so}(3)$ of $\text{SO}(3)$, with the norm $|\cdot|^2$ derived from the canonical metric on $\text{SO}(3)$. In the inference phase, the process is initialized at $O_0^s \sim \mathcal{U}(\text{SO}(3))$ and proceeds by following the geodesic in $\text{SO}(3)$, taking small steps over time $t$:

$$O_{t + \frac{1}{N}} = \exp_{O_t^s} \left( \frac{1}{N} v^{\text{ori}} \left( O_t^s, t, C^{\text{rec}} \right) \right). \quad (13)$$

**Physicchemical Features.** FM is typically applied to continuous spaces. However, certain biochemical characteristics take on discrete, categorical values. For example, each surface point can

be classified into three categories based on its hydrogen bonding potential: donor, acceptor, or neutral. This challenge also arises in protein generation tasks that focus solely on structure (Luo et al., 2022), where residue types follow a categorical distribution. To address this, previous studies (Li et al., 2024; Lin et al., 2024) have employed soft one-hot encoding to map categorical distributions to a probability simplex or directly applied FM to multinomial distributions. However, this straightforward approach may result in suboptimal performance for protein co-design. Recently, more advanced FM methods tailored for discrete spaces have been proposed (Campbell et al., 2024; Gat et al., 2024; Stark et al., 2024) to overcome these limitations.

Discrete flow models (DFMs) are generative algorithms designed to operate in discrete spaces by simulating a probability flow that transitions from noise to data. DFMs trace a trajectory of discrete variables that align the noise-to-data flow, allowing the generation of new samples. Building on the work of Campbell et al. (2024); Gat et al. (2024), we implement a DFM using CTMCs for the biochemical properties $\mathbf{\Upsilon}^s \in \mathbb{R}^{m \times 1}$. Specifically, we train a neural network $v^{\mathrm{cat}}(\cdot)$ to approximate the true denoising distribution $p_{1|t}(\mathbf{\Upsilon}_t^s \mid \mathbf{\Upsilon}_1^s)$. This is done using a cross-entropy loss, where the model learns to predict the clean data point $\mathbf{\Upsilon}_1^s$ when given a noisy input $\mathbf{\Upsilon}_t^s \sim p_{t|1}(\mathbf{\Upsilon}_t^s \mid \mathbf{\Upsilon}_1^s)$ as:

$$\mathcal{L}_{\mathrm{cat}}(\theta) = \mathbb{E}_{t \sim \mathcal{U}(0,1), p(\mathbf{\Upsilon}_1^s), p_{t|1}(\mathbf{\Upsilon}_t^s \mid \mathbf{\Upsilon}_1^s)} \left[ \log v^{\mathrm{cat}}(\mathbf{\Upsilon}_t^s, t, C^{\mathrm{rec}}) \right]. \tag{14}$$

Here, rather than a linear interpolation towards $\mathbf{\Upsilon}_1^s$ from a uniform prior $p_{t|1}^{\mathrm{unif}}(\mathbf{\Upsilon}_t^s \mid \mathbf{\Upsilon}_1^s) = \mathrm{Cat}\left(t\delta\{\mathbf{\Upsilon}_1^s, \mathbf{\Upsilon}_t^s\} + (1-t)\frac{1}{S}\right)$, we adopt an artificially introduced mask state M and the conditional path becomes (Campbell et al., 2024):

$$p_{t|1}^{\mathrm{mask}}(\mathbf{\Upsilon}_t^s \mid \mathbf{\Upsilon}_1^s) = \mathrm{Cat}(t\delta\{\mathbf{\Upsilon}_1^s, \mathbf{\Upsilon}_t^s\} + (1-t)\delta\{M, \mathbf{\Upsilon}_t^s\}). \tag{15}$$

Notably, $\mathcal{L}^{\mathrm{cat}}(\cdot)$ has a strong relation to the model loglikelihood and the Evidence Lower Bound (ELBO) used to train diffusion models (Campbell et al., 2024). It also does not depend on $R_t(\mathbf{\Upsilon}_t^s, j \mid \mathbf{\Upsilon}_1^s)$. There are many choices for $R_t(\mathbf{\Upsilon}_t^s, j \mid \mathbf{\Upsilon}_1^s)$ that all generate the same $p_{t|1}(\mathbf{\Upsilon}_t^s \mid \mathbf{\Upsilon}_1^s)$. At inference time, we pick the rate matrix for $\mathbf{\Upsilon}_t^s \neq j$ as:

$$R_t^*(\mathbf{\Upsilon}_t^s, j \mid \mathbf{\Upsilon}_1^s) := \frac{\mathrm{ReLU}\left(\partial_t p_{t|1}(j \mid \mathbf{\Upsilon}_1^s) - \partial_t p_{t|1}(\mathbf{\Upsilon}_t^s \mid \mathbf{\Upsilon}_1^s)\right)}{S \cdot p_{t|1}(\mathbf{\Upsilon}_t^s \mid \mathbf{\Upsilon}_1^s)}, \tag{16}$$

where $\mathrm{ReLU}(a) = \max(a, 0)$ and $\partial_t p_{t|1}$ can be found by differentiating our explicit form for $p_{t|1}$ in Equ. 15. This choice of $R_t^*$ assumes $p_{t|1}(\mathbf{\Upsilon}_t^s \mid \mathbf{\Upsilon}_1^s) > 0$.

**Overall Training Loss.** Combining all modalities, the final FM objective is for conditional peptide generation obtained as the weighted sum of three loss functions in Equ. 8, 12, and 14 as well as several additional loss. It can be written as:

$$\mathcal{L}_{\mathrm{CFM}} = \lambda_{\mathrm{pos}}\mathcal{L}_{\mathrm{pos}} + \lambda_{\mathrm{ori}}\mathcal{L}_{\mathrm{ori}} + \lambda_{\mathrm{cat}}\mathcal{L}_{\mathrm{cat}} + \lambda_{\mathrm{con}}\mathcal{L}_{\mathrm{con}} + \lambda_{\mathrm{str}}\mathcal{L}_{\mathrm{str}}, \tag{17}$$

where $\lambda_*$ are the hyperparameters to control the impact of different loss components. $\mathcal{L}_{\mathrm{con}}$ is the loss function for continuous biochemical properties, and $\mathcal{L}_{\mathrm{str}}$ is the FM objective for modeling the factorized distribution of residues' positions $\{a_j\}_{j=1}^{n_{\mathrm{pep}}}$, orientations $\{O_j\}_{j=1}^{n_{\mathrm{pep}}}$, amino acid types $\{x_j\}_{j=1}^{n_{\mathrm{pep}}}$, and side-chain torsion angles $\{\chi_j\}_{j=1}^{n_{\mathrm{pep}}}$ as discussed in Equ. 5 and Appendix C. The network details to parameterize the generation procedure is illustrated in Appendix A.

## 3.2 FLOW MATCHING WITH CONDTIONS

Inspired by the success of controllable image generation (Zhang et al., 2023), we propose peptide design conditioned on key factors $c$, such as sequence length $n_{\mathrm{pep}}$, cyclicity, and the presence of disulfide bonds. Our objective becomes $p(C^{\mathrm{pep}} \mid C^{\mathrm{rec}}, c)$, allowing flow models to incorporate additional conditions. Accordingly, the vector field networks are adapted to $v^{\mathrm{pos}}(x_t^s, t, C^{\mathrm{rec}}, c)$, $v^{\mathrm{ori}}(O_t^s, t, C^{\mathrm{rec}}, c)$, and $v^{\mathrm{cat}}(\mathbf{\Upsilon}_t^s, t, C^{\mathrm{rec}}, c)$. In practice, we focus on two primary goals: (1) *Cyclic peptides* offer enhanced stability by constraining the backbone, thus reducing conformational flexibility and increasing resistance to enzymatic degradation. This structure form improves binding affinity due to more defined and stable conformations. (2) *Disulfide bonds*, covalent interactions between cysteine residues, assist in proper folding and structural stabilization. These bonds protect peptides from oxidative damage and proteolytic enzymes, enhancing their resistance to harsh

Table 1: Evaluation of methods in the sequence-structure co-design task. The **best** and suboptimal results are labeled boldly and underlined.

| | Geometry | | | | Energy | | Design | |
|---|---|---|---|---|---|---|---|---|
| | AAR % ↑ | RMSD Å ↓ | SSR % ↑ | BSR % ↑ | Stb. % ↑ | Aff. % ↑ | Des. % ↑ | Div. ↑ |
| RFdiffusion (Watson et al., 2023) | 40.14 | 4.17 | 63.86 | 26.71 | **26.82** | 16.53 | **78.52** | 0.38 |
| ProteinGen (Lisanza et al., 2023) | 45.82 | 4.35 | 29.15 | 24.62 | 23.48 | 13.47 | 71.82 | 0.54 |
| Diffusion (Luo et al., 2022) | 47.04 | 3.28 | 74.89 | 49.83 | 15.34 | 17.13 | 48.54 | 0.57 |
| PepFlow (Li et al., 2024) | 51.25 | 2.07 | 83.46 | 86.89 | 18.15 | 21.37 | 65.22 | 0.42 |
| SurfFlow | **54.07** | **1.96** | **85.11** | **87.38** | 22.46 | **22.51** | 73.60 | **0.61** |

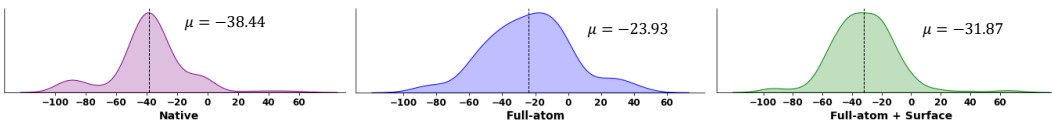

Figure 3: Binding energy distributions of designed and native peptides, where the lower is better.

industrial conditions. Additionally, disulfide bonds can improve biological activity by creating conformational constraints, further enhancing the therapeutic potential of peptide-based drugs.

# 4 EXPERIMENTS

We comprehensively evaluate SurfFlow on unconditioned and conditioned sequence-structure co-design tasks and the side-chain packing problem. For benchmarking, we use the PepMerge dataset (Li et al., 2024) derived from PepBDB (Wen et al., 2019) and Q-BioLip (Wei et al., 2024). Following the methodology of Li et al. (2024), we cluster the peptide-protein complexes based on 40% sequence identity using mmseqs2 (Steinegger & Söding, 2017), after filtering out duplicates and applying empirical criteria (*e.g.*, resolution < 4Å, peptide length between 3 and 25). This process yields 8,365 non-redundant complexes across 292 clusters. To ensure a fair and direct comparison, we employ the same test set as Li et al. (2024), consisting of 10 clusters and 158 complexes. More experimental details and additional results are elaborated on Appendix B.

## 4.1 UNCONDITIONED SEQUENCE-STRUCTURE CO-DESIGN

**Baselines.** Two lines of state-of-the-art protein design approaches are chosen as baselines. The first kind ignores the side-chain conformations, including RFDiffusion (Watson et al., 2023) and ProteinGen (Lisanza et al., 2023). RFDiffusion produces protein backbones and sequences are later forecast by ProteinMPNN (Dauparas et al., 2022). ProteinGen improves RFDiffusionby jointly sampling backbones and corresponding sequences. The other kind considers a full-atom style protein generation, including Diffusion (Luo et al., 2022) and PepFlow (Li et al., 2024).

**Evaluation Metrics.** Generated peptides are evaluated from three key aspects. (1) **Geometry**: Designed peptides should closely resemble native sequences and structures. We use the amino acid recovery rate (**AAR**) to quantify sequence identity between generated peptides and ground truth. Structural similarity is assessed through the root-mean-square deviation (**RMSD**) of $C_\alpha$ atoms after aligning the complexes. Secondary-structure similarity ratio (**SSR**) measures the proportion of shared secondary structures, while the binding site ratio (**BSR**) compares the overlap between the binding sites of the generated and native peptides on the target protein. (2) **Energy**: Our goal is to design high-affinity peptide binders that enhance the stability of protein-peptide complexes. **Affinity** is defined as the percentage of generated peptides with higher binding affinities (lower binding energies) than the native peptide, while **Stability** indicates the proportion of complexes with lower total energy than the native state. Energy calculations are performed using Rosetta (Alford et al., 2017). (3) **Design**: **Designability** reflects the consistency between designed sequences and structures. It is measured by the fraction of sequences that can fold into structures similar to their corresponding generated forms, with $C_\alpha$ RMSD < 2 Å as the threshold. We use ESMFold (Lin

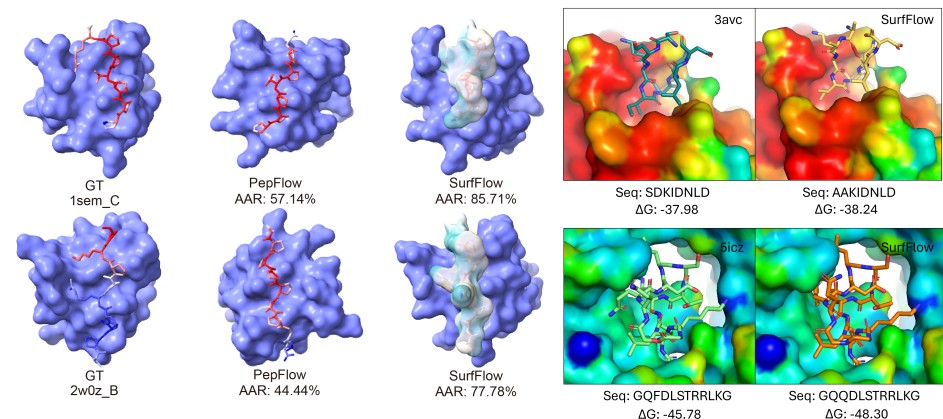

Figure 4: Peptide designed by different algorithms and references.

Figure 5: Peptide design with the cyclic condition.

et al., 2022) to refold the sequences. **Diversity**, measured as the average of one minus the pairwise TM-Score (Zhang & Skolnick, 2004), indicates structural dissimilarity among designed peptides.

**Results.** Table 1 illustrates that our SurfFlow generates significantly more diversified and consistent peptides with better binding energy and closer resembles compared to the baselines. Specifically, SurfFlow achieves the state-of-the-art AAR of 54.07% and RMSD of 1.96 Åwith improvements of 5.51% and 5.31% over the full-atom PepFlow. Besides, it also owns a stronger capacity of desiging peptides with more accurate binding ratio of 87.38% and higher affinity propertion of 22.51% (see Fig. 3). These stastics confirm the benefits of explicitly modeling the surface geometry and biochemical constraints. It is worth mentioning that the decoupled approach, RFDiffusion, attains better Stability (26.82% v.s. 22.46%) and Designability (78.52% v.s. 73.60%), as it is trained in the entire PDB and are proned towards structures with more stable motifs (Li et al., 2024). Figure 4 presents two examples of designed peptides by full-atom PepFlow and surface-based SurfFlow and Appendix D.2 shows how the positions of surface point clouds evolves over the entire time period. Evidently, SurfFlow generates peptides with topologically similar geometries, irrespective of their native length. Notably, the produced peptides share comparable side-chain compositions and conformations, enabling efficient interactions with the target protein at the appropriate binding site.

### 4.2 CONDITIONED SEQUENCE-STRUCTURE CO-DESIGN

There are several key characteristics of effective peptides widely recognized in the biological community. Cyclicity, for instance, are polypeptide chains with a circular bond sequence. Many occur naturally and exhibit antimicrobial or toxic properties, while others are laboratory-synthesized (Jensen, 2009). These cyclic peptides typically demonstrate high resistance to digestion, making them attractive to researchers developing novel oral medications like antibiotics and immunosuppressants (Craik, 2006). Additionally, evidence indicates that disulfide patterns are crucial in the folding and structural stabilization of peptides. The deliberate introduction of disulfide bridges into natural or engineered peptides can often enhance their biological activities, specificities, and stabilities (Annis et al., 1997). Given these insights, we propose a conditional co-design challenge. This challenge involves calculating the proportions of cyclic peptides and peptides containing disulfide bonds within generated peptide sets for evaluation purposes.

**Results.** Table 2 documents the results with several Key findings. Without conditional constraints, neither full-atom deep generative models nor our omni-design method met the requirements for cyclicity or disulfide bridges. Even with the incorporation of molecular surfaces, only 2-4% of the 189 designed peptides in the test samples exhibited cyclicity or contained disulfide bonds. In contrast, When trained with additional conditions and prompted to generate peptides with specific properties, SurfFlow significantly increases the proportion of peptides with desired characteristics, with cyclicity from 2.67% to 8.02% and disulfide bonds from 4.27% to 9.10%. Figure 5 visualizes two

Table 2: Proportions of cyclic peptides and peptides with disulfide bonds designed by different mechanisms and in the original PepMerge dataset.

| Metrics | PepFlow | SurfFlow (w/o $c$) | SurfFlow | PepMerge |
|---|---|---|---|---|
| **Cyclicity** | 2.13% | 2.67% | **8.02%** | 15.50% |
| **Disulfide Bond**% | 3.21% | 4.27% | **9.10%** | 18.18% |

Table 3: Evaluation results of the side-chain packing task.

| | MSE $^\circ$ $\downarrow$ | | | | Correct % $\uparrow$ |
|---|---|---|---|---|---|
| | $\chi_1$ | $\chi_2$ | $\chi_3$ | $\chi_4$ | |
| Rosseta | 38.31 | 43.23 | 53.61 | 71.67 | 57.03 |
| SCWRL4 | 30.06 | 40.40 | 49.71 | 53.79 | 60.54 |
| DLPacker | 22.44 | 35.65 | 58.53 | 61.70 | 60.91 |
| AttnPacker | 19.04 | 28.49 | 40.16 | 60.04 | 61.46 |
| DiffPack | 17.92 | 26.08 | 36.20 | 67.82 | 62.58 |
| PepFlow | 17.38 | 24.71 | 33.63 | 58.49 | 62.79 |
| SurfFlow | **17.13** | **23.86** | **31.97** | **55.08** | **63.02** |

examples with the cyclic condition. Interestingly, in the case of 3AVC, where the native peptide is non-cyclic, SurfFlow generated a novel cyclic peptide with a lower binding energy ($\Delta G = -37.98$ v.s. $\Delta G = -38.24$). For 5ICZ, which originally had a cyclic peptide, SurfFlow designed a new type of cyclic peptide with a better binding energy ($\Delta G = -45.78$ v.s. $\Delta G = -48.30$).

## 4.3 SIDE-CHAIN PACKING

Side-chain packing is a critical task in protein structure modeling, focusing on the prediction of peptide side-chain angles. Our approach generates 64 distinct side-chain conformations for each peptide using multiple models, employing a partial sampling strategy to efficiently recover the most probable side-chain angles. This method allows us to navigate the conformational space effectively while reducing computational overhead and maintaining high accuracy.

**Baselines.** We compare SurfFlow against several established approaches. These include energy-based methods such as RosettaPacker (Leman et al., 2020) and SCWRL4 (Krivov et al., 2009), which rely on physical energy minimization techniques for side-chain positioning. Additionally, we evaluate against learning-based models including DLPacker (Misiura et al., 2022), AttnPacker (McPartlon & Xu, 2023), and DiffPack (Zhang et al., 2024), which leverage various DL strategies like attention mechanisms and diffusion to predict side-chain configurations based on learned representations of protein structure.

**Metrics.** Two primary metrics are employed for evaluation. First, we calculate the Mean Absolute Error (MAE) of four key torsion angles: $\chi_1$, $\chi_2$, $\chi_3$, and $\chi_4$. Given the inherent flexibility of side chains and the importance of small deviations in structural biology, we also report the proportion of predictions that fall within a $20^\circ$ deviation from the ground truth. This additional metric captures the practical accuracy of side-chain prediction, emphasizing how well the models perform within biologically relevant error margins.

**Results.** Table 3 reports the results, it can be observed that the incorporation of molecular surfaces contributes to more accurate predictions of all four side-chain angles compared to full-atom models and other baselines. SurfFlow attains the highest correct ratio of 63.02%. The enhanced accuracy suggests that surface information provides crucial context for predicting side-chain configurations, likely by better representing the local environment and potential interactions that influence side-chain positioning.

## 5 RELATED WORKS

**Protein Design with Generative DL.** Generative models have made significant strides in protein design, particularly in applications like engineering enzyme active sites (Yeh et al., 2023). These

methods generally fall into three categories: sequence design, structure design, and co-design. In sequence design, protein sequences are crafted through techniques such as oracle-guided directed evolution (Jain et al., 2022) or by leveraging protein language models (Madani et al., 2020; Verkuil et al., 2022). Another common strategy, called fix-backbone sequence design, involves generating sequences that fit a predefined backbone structure (Ingraham et al., 2019; Jing et al., 2020; Hsu et al., 2022; Gao et al., 2022b;a; Zheng et al., 2023b). Given the importance of 3D structural information in proteins, some approaches focus on first generating protein backbone structures (Anand & Achim, 2022), which are then paired with sequence prediction models like ProteinMPNN (Dauparas et al., 2022) to determine the matching sequence. Co-design methods, which generate sequence-structure pairs simultaneously, are especially useful for antibody design (Jin et al., 2021; Kong et al., 2022; Wu & Li, 2024a). Recent studies also emphasize side-chain interactions in conditional protein generation, enabling full atomic detail (Martinkus et al., 2024; Krishna et al., 2024). However, none of these methods have addressed the simultaneous generation of protein sequence, structure, and surfaces.

From a technical standpoint, diffusion and flow-based models have become popular for designing novel and diverse proteins (Ingraham et al., 2023; Lin & AlQuraishi, 2023). Some studies focus on sequence generation alone using discrete diffusion models (Alamdari et al., 2023; Frey et al., 2023; Gruver et al., 2024; Yi et al., 2024). These models are also applied to generate protein structures in 3D or SE(3) space (Trippe et al., 2022; Anand & Achim, 2022; Bose et al., 2023; Yim et al., 2023; Wu et al., 2024). Among them, RFDiffusion (Watson et al., 2023) has seen notable success, with wet-lab validation of generated proteins. However, these methods often require a separate model for sequence generation. In co-design, earlier efforts include ProteinGenerator (Lisanza et al., 2023), which uses Euclidean diffusion over one-hot encoded amino acids while predicting structure at each step using RosettaFold (Baek et al., 2021). Protpardelle (Chu et al., 2024) applies Euclidean diffusion to structure while iteratively predicting the sequence. Multiflow (Campbell et al., 2024) introduces a DFM model over protein sequences, offering flexible conditioning during inference. Early co-design methods by (Luo et al., 2022; Shi et al., 2022) focused on designing CDR loops in antibodies. Co-design for peptides is also gaining traction, with models like PepFlow (Li et al., 2024) and PPIFlow (Lin et al., 2024) excelling in full-atom peptide design using multi-modality FM. Finally, PepGLAD (Kong et al., 2024) explored peptide structure and sequence diffusion, but no code has been made available.

**Molecular Surface Modeling.** The properties of a protein's molecular surface are crucial in determining the nature and strength of its interactions with other molecules. This surface is shaped by van der Waals (vdW) radii (Connolly, 1983) and is often represented as meshes created from signed distance functions. MaSIF (Gainza et al., 2020) was a pioneering effort in applying mesh-based geometric deep learning to abstract internal protein folds and study protein interactions. Later, Sverrisson et al. (2021) simplified the process by representing molecular surfaces as point clouds, assigning atom types to each point to reduce pre-computation costs. Other key works have integrated protein surface data with structural information in a multimodal approach (Somnath et al., 2021), using advanced pretraining techniques (Wu & Li, 2024b) and implicit neural representations (INRs)(Park et al., 2019) for self-supervised learningLee et al. (2023) and dynamic structure modeling Sun et al. (2024). Despite these developments, protein design based on surface characteristics remains relatively unexplored. However, recent progress, such as Gainza et al. (2023)'s extension of MaSIF for *de novo* binder design, and SurfPro (Song et al., 2024), which eliminates the need for handcrafted feature extraction, has begun to address this gap by generating functional proteins directly from surface data. SurfFlow stands out as the first approach to generate all protein modalities simultaneously using flow-based algorithms.

## 6 CONCLUSION

This work presents SurfFlow, a novel deep generative model that produces all protein modalities – sequence, structure, and surface – concurrently. We apply SurfFlow to solving a specific peptide design challenge and integrate some key characteristics like cyclicity and disulfide bonds into the generation process. Empirical results prove the reasonability and promise of considering molecular surfaces for protein discovery. Limitation and future work is elucidated in Appendix E.

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

## A   Parameterization with Networks

In order to model the joint distribution of the peptide based on its target protein $p\left(C^{\text{pep}} \mid C^{\text{rec}}\right)$, we adopt an encoder-decoder framework to generate peptides. The encoder extracts the geometric and biochemical features of the receptor $C^{\text{rec}}$ as the condition of the generation process, while the decoder regresses the vector fields of our multi-modality flow matching architecture.

**Encoder.**   We first utilize a time-independent equivariant geometric encoder to capture the context information of the receptor. Specifically, it takes the sequence and structure of the target protein $C_{\text{rec}}$ and computes the hidden residue representations $\boldsymbol{h}_{\text{rec}} \in \mathbb{R}^{n_{\text{rec}} \times \psi_{\text{rec}}}$ and the residue-pair embeddings $\boldsymbol{z}_{\text{rec}} \in \mathbb{R}^{n_{\text{rec}} \times \psi_{\text{pair}}}$.

**Decoder.**   The decoder receives $(\boldsymbol{h}_{\text{rec}}, \boldsymbol{z}_{\text{rec}})$ and is time-dependent. It consists of two geometric networks: one is the 6-layer Invariant Point Attention (IPA) module (Jumper et al., 2021) to regress the vector fields of the internal structures $\{(a_j, O_j, \boldsymbol{x}_j, \boldsymbol{\chi}_j)\}_{j=1}^{n_{\text{pep}}}$, and the other is an variant of equivariant graph neural networks (EGNN) (Satorras et al., 2021) to regress the vector fields of the surface geometry $\{(\boldsymbol{x}_i^s, \boldsymbol{n}_i^s, \boldsymbol{\tau}_i^s, \boldsymbol{\Upsilon}_i^s)\}_{i=1}^{m}$. To be specifica, we first sample a random timestep $t \sim \mathcal{U}(0,1)$ to construct the time-dependent vector fields for every modality of the peptide $C_{\text{pep}}$, containing sequence, structure, and surface. Both IPA and EGNN take the timestamp $t$, the interplant state of peptide's internal structure $(a_t, O_t, \boldsymbol{x}_t, \boldsymbol{\chi}_t)$ as well as receptor's information $(\boldsymbol{h}_{\text{rec}}, \boldsymbol{z}_{\text{rec}})$ as input, and the interplant state of peptide's surface $(\boldsymbol{x}_t^s, \boldsymbol{n}_t^s, \boldsymbol{\tau}_t^s, \boldsymbol{\Upsilon}_t^s)$ is also forwarded into EGNN. Subsequently, IPA recovers the internal structure of original peptide $\left(\hat{a}_1, \hat{O}_1, \hat{\boldsymbol{x}}_1, \hat{\boldsymbol{\chi}}_1\right)$, while the 3-layer EGNN recovers the surface $\left(\hat{\boldsymbol{x}}_1^s, \hat{\boldsymbol{n}}_1^s, \hat{\boldsymbol{\tau}}_1^s, \hat{\boldsymbol{\Upsilon}}_1^s\right)$. Moreover, two additional losses containing the backbone position loss $\mathcal{L}_{\text{bb}}(\theta)$ and the torsion angle loss $\mathcal{L}_{\text{tor}}(\theta)$ (Li et al., 2024) is imposed for extra constraint.

**Equivariance.**   The joint distribution $p\left(C^{\text{pep}} \mid C^{\text{rec}}\right)$ must satisfy the roto-translational equivariance to ensure the generalization. That is, for any translation vector $\boldsymbol{\epsilon} \in \mathbb{R}^3$ and for any orthogonal matrix $O \in \mathbb{R}^{3 \times 3}$, it should satisfy:

$$p\left(OC^{\text{pep}} + \boldsymbol{\epsilon} \mid OC^{\text{rec}} + \boldsymbol{\epsilon}\right), \tag{18}$$

where $OC^{\text{pep}} + \boldsymbol{\epsilon} = \{(a_j, O_j, O\boldsymbol{x}_j + \boldsymbol{\epsilon}, \boldsymbol{\chi}_j)\}_{j=1}^{n_{\text{pep}}} \cup \{(O\boldsymbol{x}_i^s + \boldsymbol{\epsilon}, \boldsymbol{n}_i^s, \boldsymbol{\tau}_i^s, \boldsymbol{\Upsilon}_i^s)\}_{i=1}^{m}$. Following the standard operation called zero-mass-center (Yim et al., 2023; Lin et al., 2024), we substract the mass center of the receptor from all inputs' coordinates to achieve the invariance to translation, which also improves the training stability. Moreover, it can be proven that when the prior distributions $p(a_0), p(O_0), p(\boldsymbol{x}_0), p(\boldsymbol{\chi}_0), p(\boldsymbol{x}_0^s), p(\boldsymbol{n}_0^s), p(\boldsymbol{\tau}_0^s)$, and $p(\boldsymbol{\Upsilon}_0^s)$ are SE(3)-invariant, while the vector fields $v^{\text{cat}}(\cdot)$ and $v^{\text{con}}(\cdot)$ are SE(3)-invariant, and the vector field $v^{\text{ori}}(\cdot)$ is SO(3)-equivariant and T(3)-invariant, and the vector field $v^{\text{pos}}(\cdot)$ is SE(3)-equivariant, then the density $p\left(C^{\text{pep}} \mid C^{\text{rec}}\right)$ generated by the ODE sampling process is SE(3)-equivariant. Notably, the choice of IPA and EGNN guarantees the equivariance and invariance requirement of those vector fields.

**Conditional Design.**   Controlling the output of deep generative models such as diffusions or flows has become a hotspot in recent years. Apart from the receptor $C_{\text{rec}}$, we often want to create peptides with specific conditions, such as cyclicity and disulfide bonds. Conventionally, conditional controls involve classifier guidance (Dhariwal & Nichol, 2021) and classifier-free guidance (Ho

et al., 2020). Evidence indicates that with enough high-quality training data, classifier-free guidance tends to yield better results, being able to generate an almost infinite number of sample categories without the need to retrain a classifier architecture. This leads to wide usage of classifier-free guidance in modern AI systems (Ramesh et al., 2022; Zheng et al., 2023a). Due to this superiority, we adopt the classifier-free guidance to realize the insertion of peptide conditions $c$ and transfer $p(C_{\mathrm{pep}}|C_{\mathrm{rec}})$ to $p(C_{\mathrm{pep}}|C_{\mathrm{rec}}, c)$. Specifically, we denote the null condition by $\emptyset$ by convention and set $p(C_{\mathrm{pep}}|C_{\mathrm{rec}}, \emptyset) := p(C_{\mathrm{pep}}|C_{\mathrm{rec}})$ and $u_t(\cdot|\cdot, \emptyset) := u_t(\cdot|\cdot)$. Then taking the surface point positions for example, our training loss becomes (Zheng et al., 2023a):

$$\mathcal{L}_{\mathrm{pos}}(\theta) = \mathbb{E}_{t \sim \mathcal{U}(0,1), b, p(\boldsymbol{x}_1^s), p(\boldsymbol{x}_0^s), p(\boldsymbol{x}_t^s|\boldsymbol{x}_0^s, \boldsymbol{x}_1^s)} \left\| v^{\mathrm{pos}}\left(\boldsymbol{x}_t^s, t, C^{\mathrm{rec}}|(1-b) \cdot c + b \cdot \emptyset\right) - (\boldsymbol{x}_1^s - \boldsymbol{x}_0^s) \right\|_2^2,$$

(19)

where $b \sim \mathrm{Bernoulli}(p_{\mathrm{uncond}})$ indicates the probability to use the null condition.

## B EXPERIMENTAL DETAILS

### B.1 TRAINING AND SAMPLING

All experiments are implemented on 4 NVIDIA A100 GPUs. Specifically, we train SurfFlow for 320K iterations and set the initial learning rate of 5e-4. A plateau cheduler is used with a factor of 0.8 and patience of 10. The minimum learning rate is 5e-6. The batch size was 32 for each distributed node. An Adam optimizer is used with a gradient clipping. A dropout ratio of 0.15 is adopted for the EGNN decoder. The weights for different loss components are set as $\lambda_{\mathrm{pos}} = 0.2$, $\lambda_{\mathrm{ori}} = 0.2$, $\lambda_{\mathrm{cat}} = 1.0$, $\lambda_{\mathrm{con}} = 1.0$, and $\lambda_{\mathrm{str}} = 1.0$. For the encoder part, the residue embedding size is set as $\psi_{\mathrm{rec}} = \psi_{\mathrm{pair}} = 128$. For the decoder part, the node and edge embedding sizes are set as 128 and 64 for IPA, respectively, while as 16 and 8 for EGNN, respectively, since the number of surface points $m + m'$ are orders of magnitude larger than the number of complex's residues $n_{\mathrm{pep}} + n_{\mathrm{rec}}$. We set the length of generated peptides the same as the length of their corresponding native peptides. We download the PepMerge data (Li et al., 2024) from its official repository: `https://drive.google.com/drive/folders/1bHaKDF3uCDPtfsihjZsOzmjwF6UU1uVl`.

### B.2 PHYSICHEMICAL SURFACE FEATURES

Three types of surface biochemical properties are leveraged in the experiments. To be specific, free electrons and potential hydrogen bond donors (FEPH) is a categorical variable, while electrostatics and hydropathy are continuous variables. Therefore, $\psi_\tau = 1$ and $\psi_\Upsilon = 2$. We resort to MaSIF (Gainza et al., 2020)'s scripts to acquire these surface features.

**Free electrons and proton donors.**    The location of FEPH in the molecular surface was computed using a hydrogen bond potential as a reference. Vertices in the molecular surface whose closest atom is a polar hydrogen, a nitrogen, or an oxygen were considered potential donors or acceptors in hydrogen bonds. Then, a value from a Gaussian distribution was assigned to each vertex depending on the orientation between the heavy atoms. These initial values range from -1 (optimal position for a hydrogen bond acceptor) to +1 (optimal position for a hydrogen bond donor). Then the point is determined as an acceptor or a donor (a binary label) by whether FEPH is negative or positive.

**Hydropathy.**    Each vertex was assigned a hydropathy scalar value according to the Kyte and Doolittle scale of the amino acid identity of the atom closest to the vertex. These values, in the original scale, ranged between -4.5 (hydrophilic) to +4.5 (most hydrophobic) and were then normalized to be between -1 and 1.

**Poisson-Boltzmann continuum electrostatics.**    PDB2PQR was used to prepare protein files for electrostatic calculations and APBS (v.1.5) was used to compute Poisson-Boltzmann electrostatics for each protein. The corresponding charge at each vertex of the meshed surface was assigned using Multivalue, provided within the APBS suite. Charge values above +30 and below -30 were capped at those values and then values were normalized between -1 and 1.

## C  Flow Matching for Internal Structures

$\mathcal{L}_{\text{str}}(\theta)$ in Equ. 17 computes the loss for peptide's structures $p\left(\{(a_j, O_j, \boldsymbol{x}_j, \boldsymbol{\chi}_j)\}_{j=1}^{n_{\text{pep}}} \mid C^{\text{rec}}\right)$.
Here, we provide a quick view of how to conduct full-atom CFM for residues' positions $\boldsymbol{x}$, types $a$, backbone torsions $O$, and side-chain angles $\boldsymbol{\chi}$. To be concrete, the objective is $\phi_t\left(\boldsymbol{x}_0, \boldsymbol{x}_1\right) = t\boldsymbol{x}_1 + (1-t)\boldsymbol{x}_0$ for residues' positions $\boldsymbol{x}$, $\exp_{O_0}\left(t\log_{O_0}\left(O_1\right)\right)$ for backbone orientations $O$, $\phi_t\left(\boldsymbol{\chi}_0, \boldsymbol{\chi}_1\right) = [t\boldsymbol{\chi}_1 + (1-t)\boldsymbol{\chi}_0] \mod 2\pi$ for side-chain angles $\boldsymbol{\chi}$, and $\phi_t\left(\tilde{a}_0, \tilde{a}_1\right) = t\tilde{a}_1 + (1-t)\tilde{a}_0$ for residue types $a$, where $\tilde{a}$ is the representation of $a$ using a soft one-hot encodeing operation and satisfies $\text{logit}(a_i) = \tilde{a}_i \in \mathbb{R}^{20}$.

## D  Additional Results and Visualization

### D.1  Ablation Studies

We conduct experiments to investigate the contributions of each component of our SurfFlow model. Table 4 shows that the removal of the biophysical features significantly reduces the performance with a drop of 3.76% in Designability and 4.91% in Diversity. Besides, it also indicates that inclusion of surface orientation is beneficial, which brings an improvement of 1.94% in AAR.

Table 4: Ablation studies on different components of SurfFlow.

|  | Geometry | | | | Energy | | Design | |
|---|---|---|---|---|---|---|---|---|
|  | AAR % ↑ | RMSD Å ↓ | SSR % ↑ | BSR % ↑ | Stb. % ↑ | Aff. % ↑ | Des. % ↑ | Div. ↑ |
| SurfFlow | **54.07** | **1.96** | **85.11** | **87.38** | **22.46** | **22.51** | **73.60** | **0.61** |
| w/o Position | 53.26 | 1.99 | 84.79 | 87.15 | 21.30 | 22.38 | 72.09 | 0.60 |
| w/o Orientation | 53.04 | 2.00 | 84.60 | 87.04 | 20.79 | 22.46 | 72.36 | 0.60 |
| w/o Biophysical Prop. | 52.31 | 2.03 | 83.96 | 86.98 | 19.55 | 22.47 | 70.83 | 0.58 |

### D.2  Surface Point Cloud Evolution

We give some examples of how the surface point clouds move from time $t = 0$ to the terminal time $t = 1$. It can be seen that after time $t = 0.5$, those clouds begin to take shapes.

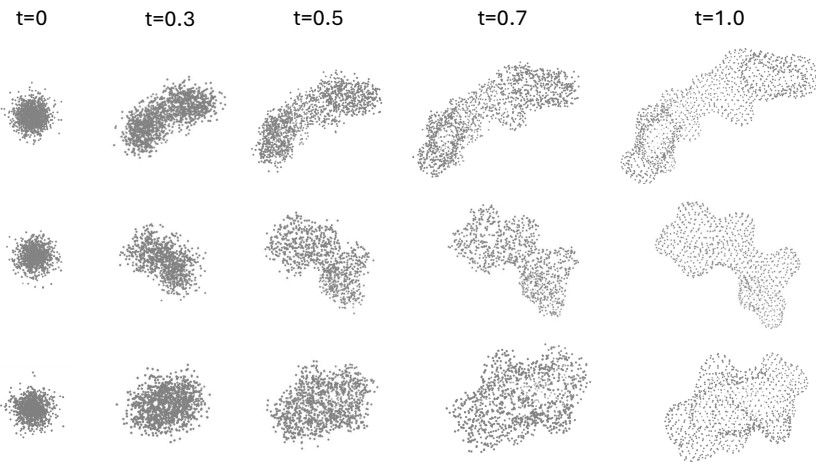

Figure 6: Visualization of the evolution of surface points' positions over time $[0, 1]$.

# E  LIMITATIONS AND FUTURE WORKS

Despite the enhancement of our SurfFlow over the original full-atom design mechanism, there are still rooms for future explorations. For instance, further improvements can be expected if the surface information of the receptor's surface information is considered and incorporated into the joint distribution modeling. Namely, our objective becomes $C_{\text{rec}} = \{a_j, O_j, \boldsymbol{x}_j, \boldsymbol{\chi}_j\}_{j=1}^{n_{\text{rec}}} \cup \{\boldsymbol{x}_i^s, \boldsymbol{n}_i^s, \boldsymbol{\tau}_i^s, \boldsymbol{\Upsilon}_i^s\}_{i=1}^{m'}$, where $m'$ is the number of receptor's surface points. Moreover, the sucess of RFDiffusion implies that pretraining on regular proteins in PDB can be benefitial.

