# OpenReview forum: "Surface-based Peptide Design with Multi-modal Flow Matching"
_ICLR.cc/2025/Conference — Submitted to ICLR 2025_

### Official Review · Reviewer_cG9z · 2024-10-26

**Soundness:** 2
**Presentation:** 2
**Contribution:** 2
**Rating:** 5
**Confidence:** 3

**Summary:**

In this paper, the authors proposed a multi-modality conditional flow-matching method for peptide design. On PepMerge benchmark, SurfFlow consistently outperforms full-atom baselines.

**Strengths:**

1. The paper is quite clear and the code is provided.
2. The authors compared SurfFlow with extensive baselines including RFDiffusion and PepFlow.

**Weaknesses:**

1. The technical contribution is limited as PepFlow and PpFlow already apply multi-modality flow matching to peptide generation.
2. The description of how to model and leverage surface property is quite limited. Therefore, the reviewer fails to observe the key difference with previous methods.
3. The authors should put the ablation studies into the main paper to see which component contribute to the performance improvement.

**Questions:**

N.A.

---

### Official Review · Reviewer_9wEt · 2024-10-29

**Soundness:** 3
**Presentation:** 3
**Contribution:** 2
**Rating:** 5
**Confidence:** 3

**Summary:**

This work presents SurfFlow, a novel deep generative model that produces protein sequence, structure, and surface, concurrently. SurfFlow conconsistently outperforms full-atom baselines across all metrics in the PepMerge benchmark.

**Strengths:**

- Full-atom peptide design is an important problem.
- Surface is rarely considered by previous full-atom methods.
- This paper applies Discrete flow models to process categorical physichemical features.

**Weaknesses:**

- For surface generation, SurfFlow seems to combine several existing flow-matching algorithms.
- Ablation studies are needed to verify the design choice. The main contribution is to incorporate surface. So, what's the difference between other atom-based models? Have you tried other ways to process the surface?
- Why co-design methods always underperform RFDiffusion on Energy and Design. Can you think of ways to improve on this?

**Questions:**

See Weaknesses

---

### Official Review · Reviewer_ucuH · 2024-11-01

**Soundness:** 3
**Presentation:** 2
**Contribution:** 2
**Rating:** 5
**Confidence:** 5

**Summary:**

This paper proposes a flow matching-based generative model for peptide design that incorporates surface features and simultaneously generates three modalities: sequence, geometry, and surface. The model demonstrates improved performance across various metrics compared to previous approaches.

**Strengths:**

The study uses a rich set of evaluation criteria, including sequence recovery, energy, stability, and design ability, providing a thorough assessment of the model's performance.

**Weaknesses:**

## Insufficient Detail on Peptide Amino Acid Determination
1. The process of obtaining the final peptide amino acids requires more detailed explanation.
2. The method for determining peptide sequence length is not clearly described.

## Lack of Computational Efficiency Comparison
1. The paper does not provide a comparison of overall processing times between methods.
2. Surface feature calculation may introduce significant computational overhead. A time comparison with methods like RFDiffusion, which doesn't require such CPU-intensive workloads, would better illustrate the trade-offs.

## Surface Consistency Concerns
1. The model seems to generate surface positions and corresponding physico-chemical features simultaneously. While structure consistency is considered in the metrics, it's unclear how the generated surface of generated protein-peptide structures is similar with the calculated surface of the generated protein-peptide structures.
2. If significant discrepancies exist, the necessity of surface generation in the model should be justified.

## Limited Ablation Study
1. The purpose of the ablation study (whether it focuses on generation loss or input features) is not clearly specified.
2. If the impact of surface generation is minimal, the primary contribution of this research may be limited to the addition of surface features.

## Recovery vs Stability
1. Because surface is mostly determined the functional groups, sequence recovery would be higher and corresponding geometric metrics would be also increased. Beside geometric features, the model is not pretty strong compared to RFDiffusion, also, unstable peptides are useless because peptides are already unstable, so, lower stability arises the questions of use on real world discovery.

**Questions:**

## Lack of Computational Efficiency Comparison
1. Can you provide speed analysis of overall design protocol?

## Surface Consistency Concerns
1. Please analyze the surface consistency between the generated surfaces and calculated surfaces.

## Limited Ablation Study
1. Can you provide ablation studies on input features and generation loss from surfaces?

## Recovery vs Stability
1. Can you provide the reason of why your model is relatively week on stable peptide generation?

---

### Official Review · Reviewer_P617 · 2024-11-03

**Soundness:** 3
**Presentation:** 3
**Contribution:** 2
**Rating:** 3
**Confidence:** 3

**Summary:**

The authors introduce “omni-design” of peptides, which is the simultaneous generation of the peptide sequence, structure, and surface. Omni-design is motivated by the fact that the function of a peptide is determined by its structure and surface characteristics. To design peptides in this paradigm, the authors introduce SurfFlow, a multi-modal conditional flow matching model that generates a peptide sequence, structure, and surface all conditioned on a protein pocket. The experiments show that SurfFlow generates peptides with better binding and similarity to known ligands than baselines.

**Strengths:**

* Relatively clear presentation of the proposed method, and extensive background given on the types of flow matching models used
* Reasonable design choices given the proposed problem setting
* Comprehensive set of recent baselines, and strong experimental results on a set of multiple important properties (binding energy and similarity to known ligands)
* Explicit consideration of peptide cycles and disulfide bonds, which are important real-world considerations

**Weaknesses:**

My primary concern is the lack of biological and chemical motivation behind many of the design choices:

* The central novelty of the paper seems to be the consideration of the peptide surface in the generation process. However, the surface of a peptide is determined entirely by its amino acid sequence. Therefore, what is the motivation behind jointly generating both the amino acid sequence and surface shape? It is not possible to modify the surface shape of a peptide beyond what is determined by the amino acid sequence, so what is the purpose of generating a surface if it cannot be used?
* Similarly to above, the physicochemical properties of the surface are entirely determined by the amino acid sequence. Therefore, why model the amino acid sequence and physicochemical properties separately? Additionally, the surface properties of all amino acids are already known (such as charge, hydrophobicity, etc.), so why generate these properties instead of just using their known values?
* What is the meaning of the vector orientations on the surface? The “surface” of a peptide is really just an arbitrary boundary determined by electron density, and so there is no such thing as a directionality to the surface. While the authors distinguish between “concave” and “convex” parts of the surface, I’m not sure how that is relevant or worth modeling. And again, it is impossible for a biologist to somehow synthesize a peptide with a different surface orientation than what is determined by the amino acid sequence.
* I disagree that the physicochemical characteristics considered in the paper should be represented with categorical values. Specifically, the authors treat hydrogen bonding as a categorical variable (acceptor/donor/neutral), but I think it would be more suitable to consider hydrogen bonding potential as a continuous variable. Hydrogen bonds are a special type of dipole-dipole interaction, and even among hydrogen bond donors there can be different strengths. Therefore, why not consider hydrogen bonding as a continuous variable?

Taken together, it is hard for me to understand why the proposed model achieves superior results compared to models that only consider the sequence. If the authors successfully motivate their design choices from a biochemical viewpoint, I would be happy to raise my score. However, as it stands, it is hard to see how any output of the model besides the amino acid sequence is useful for biologists. Why not just model the amino acid sequence only?

**Questions:**

* “Full-atom” is usually used to refer to specifying the individual atoms in a molecule. However, the paper only considers the residue level, so why is the phrase “full-atom” used?
* How is the L_con loss calculated?
* What are the baselines used in Figure 3, what do they mean, and how are they trained?
* Are all baselines trained on the same dataset? If so, why is it stated that RFDiffusion is trained on the PDB? If not, how is the comparison fair?

---

### Official Review · Reviewer_LcGN · 2024-11-04

**Soundness:** 3
**Presentation:** 3
**Contribution:** 3
**Rating:** 5
**Confidence:** 2

**Summary:**

1. This paper tackles the problem of conditional protein (peptide) generation by leveraging flow matching. It proposes to go beyond peptide sequence and structure and additionally model peptide surface and hence introduce an omni-design peptide generative workflow.
2. The authors perform a full atom conditional flow matching (CFM) for peptide's structure to to learn residues' position, residue identities, backbone torsion, and side chain angles. Concurrently, they perform flow matching to learn peptide surface. The position, normal vector and continious physiochemical properties of surface point cloud are learned through CFM while they utilize discrete flow matching to learn the discrete physiochemical features.
3. The paper extends its method for further conditioning on cyclicity and disulphite bonds and additionally show results for side-chain packing task.

**Strengths:**

1. The manuscript is very well written and has broken down complex flow matching objectives into clear and digestible pieces. It was relatively to follow through most of their method.
   2. The use of peptide surface in conjunction to sequence and structure is novel and empricial results seem to point at the benefits of using them

**Weaknesses:**

1. The key novelty  of this work, beyond the usage of surface should be pointed out in the introduction. A few sections (such as Euclidean CFN for postion and physiochemical features susection - more on this to follow) appear to be a straight-foward adaptation of earlier works. Hence, the contributions of this work may be missed by the readers if not explicity pointed out. I had to extensively read through the appendix to appreciate the technical contributions of this work as it was not very evident from the main text. For instance how is the physiochemical features different from Campbell et. al 2024?
2. L$_{con}$ in eq 17 should be clarified. It is not mentioned explicity in the paper.
3. Which physiochemical propeties for surface are used should be mentioned in main text or the reference to it in appendix should be provided.
4. A key consideration of protein generative models are their training and sampling time performance. It would be beneficial to see wall-times of the proposed method and competing state-of-the-art methods.
5. Since the key contribution is the addition of peptide surface, justifying its usage is critical and to back it up with experimental results on its utility is a must. I couldn't find experiments to ablate all peptide surface component in its entirety and fair comparision to other seq-structure codesign methods. To my understanding, the ablation results in appendix remove 1 sub-component within "surface design" component but not the entire "surface design" component.
6. Minor:
 - L59; lock-and-key and induced fit can use references.
 - details of v$^{pos}$, v$^{ori}$, v$^{cat}$ networks should referenced in main text as these annotations are not directly used in the appendix.

**Questions:**

- What is the benefit of modeling both $p((.)^{pep})$ and physiochemical properties (and even the position and orientation) of the surface. If the peptide is considered at its atom level resolution, wouldn't the coordinates, position, identity etc. of the atoms sufficiently characterize the peptide surface?
 And as an extension, in appendix C, authors mention 'full-atom' CFN of 'residues'. The full atom specifications seem to be misleading.
 - On the AAR metric, for any given $C^{rec}$, there can be multiple good binding peptides. How does the sequence identity to the ground truth ensures that the most suitable peptide binder is generated? Does the training dataset provide a guarantee on suitability of the binding peptide and the binding strength?

---

### Author Response · Authors · 2024-11-26

We sincerely thank all five reviewers for your thoughtful reviews and valuable feedback on our submission. We greatly appreciate the time and effort you dedicated to evaluating our work and providing constructive suggestions. Your insights have highlighted several areas for improvement, which we believe will significantly enhance the quality of our research.

After careful consideration, we have decided to focus our efforts on addressing your suggestions and further refining our manuscript for submission to a later conference. As such, we will not be submitting a rebuttal during this period.

Thank you again for your invaluable feedback and for contributing to the ongoing improvement of our work. We look forward to integrating your recommendations and sharing an improved version of our study in the future.

Warm regards,

---

### Meta-Review · Area_Chair_LUZh · 2024-12-22

**Metareview:**

The paper considers surface as an important component when modeling new therapeutic peptides and proposes a flow matching model to generate proteins’ sequence, structure, and surface conditioned on an intended receiving pocket. They empirically show it can generate peptides with better binding qualities on the PepMerge benchmark compared to existing baselines.

The reviewers acknowledged the motivation of the work in including surface in a multi-modal generative modeling, the relative clarity of the writing and presentation, and the comprehensive set of baselines,

At the same time, they raised several concerns regarding the technical novelty, lack of targeted experiments to give insight on the inclusion of surface in modelling and generally absence of the necessary ablation studies, lack of proper motivation and detail on the included physicochemical properties and how they are encoded, and a necessity on computational time comparison with other approaches for both training and sampling.

Therefore, the reviewers unanimously lean towards rejection. The authors appreciated the feedback and mentioned their intention to submit to another venue instead of a rebuttal.

The AC recommends rejection.

**Additional Comments On Reviewer Discussion:**

The paper received feedback from five expert reviewers both from machine learning and bioinformatics. They all leaned towards rejection.

---

### Decision · Program_Chairs · 2025-01-22

Reject